# The relationship between wellness and training and match load in professional male soccer players

**Rafael Franco Soares Oliveira**[1,2,3]*, **Rui Canário-Lemos**[4,5], **Rafael Peixoto**[4,5], **José Vilaça-Alves**[3,4,5], **Ryland Morgans**[6], **João Paulo Brito**[1,2,3]

**1** Sports Science School of Rio Maior–Polytechnic Institute of Santarém, Rio Maior, Portugal, **2** Life Quality Research Centre, Rio Maior, Portugal, **3** Health Sciences and Human Development (CIDESD), The Research Centre in Sports Sciences, Vila Real, Portugal, **4** University of Trás-os-Montes e Alto Douro, Vila Real, Portugal, **5** Research Group in Strength Training and Fitness Activities, GEETFAA, Vila Real, Portugal, **6** Football Performance Hub, University of Central Lancashire, Preston, United Kingdom

* rafaeloliveira@esdrm.ipsantarem.pt

**Data Availability Statement:** Data cannot be shared publicly because of data protection law from 25th may, 2018 in Portugal. Moreover, the club from where data was collected does not

## Abstract

The aims of this study were to: (i) analyse the within-microcycle variations in professional soccer players; (ii) analyse the relationships between wellness and training and match load demands; (iii) analyse the relationships between match-day (MD) demands and wellness during the following day (MD+1); and (iv) analyse the relationships between MD and wellness during the day before match-play (MD-1). Thirteen professional soccer players (age: 24.85±3.13 years) were monitored daily over 16-weeks for wellness and training and match-play intensity. The daily wellness measures included fatigue, quality of sleep, muscle soreness, mood and stress using a 1–5 scale. Internal intensity was subjectively measured daily using the rating of perceived exertion (RPE) and the multiplication of RPE by session duration (s-RPE). While external intensity was quantified utilising high-speed running, sprinting, and acceleration and deceleration metrics. Data was analysed from each training session before (i.e., MD-5) or after the match (i.e., MD+1). Repeated measures ANOVA or Friedman ANOVA was used to analyse the aims (i) where Spearman correlation was applied to analyse the relationships between the aims (ii) and (iii) between sleep quality and training intensity. The main results for aim (i) showed that MD+1 presented the lowest values for wellness variables (p < 0.05). While MD-1 presented the lowest internal and external load values (for all variables), with MD presenting the highest values (p < 0.05). Regarding aim (ii), the main result showed significant large negative correlations between fatigue and s-RPE (r = -0.593; p = 0.033). Considering aim (iii), significant small to very large negative correlations were found for sleep quality, fatigue and muscle soreness with all internal and external variables (p < 0.05). Lastly, the main results for aim (iv) showed large negative correlations for fatigue and session duration; fatigue and s-RPE; muscle soreness and session duration; muscle soreness and s-RPE; and muscle soreness and decelerations (p < 0.05, for all). The main conclusions were that MD had an influence on wellness and internal and external training intensity, notably MD-1 and MD+1 were most affected. In this regard, a tendency of higher

approve data sharing due to sensitive participant information. Interested researchers may contact the corresponding author (Rafael Oliveira, rafaeloliveira@esdrm.ipsantarem.pt) and Life Quality Research Centre, TELEPHONE: +351 243 999 280. https://www.cieqv.pt/ (cieqv.geral@gmail.com).

**Funding:** this research was funded by the Portuguese Foundation for Science and Technology, I.P., Grant/Award Number UIDP/04748/2020, but the funders had no role in the design of the study; in the collection, analyses, or interpretation of data; in the writing of the manuscript, or in the decision to publish the results.

**Competing interests:** The authors have declared that no competing interests exist.

internal and external intensity on MD was associated with lower wellness measures of sleep quality, muscle soreness and fatigue on MD+1.

## Introduction

The quantification of training and match demands on soccer players is common practice to reduce injury risk and overtraining [1–3]. For instance, a recent study showed that coaches and practitioners employ periodisation strategies by adjusting training intensity considering the previous session [4]. In this sense, athlete monitorisation includes the quantification of training/match demands (e.g., locomotor/mechanical and psychophysiological), and player wellness and readiness [5]. On the one hand, psychophysiological demands are associated with internal intensity monitoring using subjective or objective measures (e.g. rating of perceived exertion–RPE and heart rate, respectively) [2,6], while locomotor/mechanical demands are associated with external intensity monitoring using global positioning system (GPS) variables (e.g., distances covered at various running speed or accelerations). Wellness is usually measured via questionnaires as previously proposed by Hooper and Mackinnon [7] and McLean et al. [8]. These questionnaires include different items such as fatigue, quality of sleep, muscle soreness, mood and stress [8].

Considering the existing literature, wellness can be related to training or match-play intensity and thus can represent how players respond to different levels of intensity. Specifically, it has been suggested that poor perceptual wellness and high training intensity values should be considered when amending intensity, while higher levels of intensity followed by a positive level of perceptual wellness may be a positive sign to continue the current training and intensity process [5].

In this sense, some studies examined the interaction between training intensity and wellness variables in professional [9–11], and youth [12–15] soccer. While direct relations between wellness variables and training intensity have revealed small-to-moderate magnitudes of correlation [16], it has been found large magnitudes of correlations between well-being outcomes and some measures that identify accumulated training intensities and variability of these demands [12].

Previous data has suggested that other factors may also influence the direction and magnitude of correlations. For instance, training intensity is collected during the training or match session, while wellness is usually collected prior to the training session or match, which may suggest that this subjective wellness value includes not only the current wellness perception, but the perception of the day before. This was evident in a recent study in youth soccer players that analysed such relationships, although only sleep and intensity variables were investigated [15]. Specifically, this study showed that higher intensity sessions contributed to improved sleep and that longer sleep duration contributed to higher session-RPE (s-RPE) values [15]. Considering that microcycle demands may vary within each week [17], data on the relationships between intensity and other wellness variables such as fatigue, muscle soreness, mood and stress is also required to provide useful insights for coaches [18,19].

Therefore, the aims of this study were to: (i) analyse the within-microcycle variations in professional soccer players; (ii) analyse the relationships between wellness and training and match load demands; (iii) analyse the relationships between match-day (MD) demands and wellness during the following day (MD+1); and (iv) analyse the relationships between MD and wellness during the day before match-play (MD-1). Considering previous studies [9–15], it was hypothesised that intensity and wellness vary across a typical professional soccer

microcycle and that relationships between intensity and wellness variables are evident. Furthermore, correlations between MD+1 wellness and match demands and MD-1wellness and match demands are expected based on existing research [15].

## Materials and methods

### Design

An observational study design of the first 16-week period of the in-season period (July to November) following a similar design of previous studies [14,15] was conducted. The players were monitored daily for sleep quality, fatigue, muscle soreness, stress, mood, session duration and training/match intensity. During the analysed period, 70 training sessions (usually performed at 10.00 a.m.) and 15 official matches were performed. Training and match data were collected at the soccer club's outdoor training pitches by staff members who were also researchers of the present study. The present investigation did not influence or alter any training session design planned by the coach.

All microcycles were organised according to the following match, thus all microcycles included only one match. Consequently, there were different microcycle schedules (Table 1).

### Participants

Thirteen professional male soccer players (age: 24.85±3.13 years; body mass: 71±6.8 kg; body height: 178.2±6.9 cm; fat mass: 8.52±1.16%; professional experience: 7.07±2.75 years) participated in the current study as a convenience sample (non-probability sampling method). Players belonged to a European soccer team that played in the first division of its national league. The eligibility criteria for participant inclusion were: (i) completing 90% of the total number of training sessions (full session duration); (ii) completed every wellness and training intensity report over the data collection period. All information was collected daily across the study

**Table 1. Number of training session per microcycle.**

| Microcycle | Training sessions per microcycle |
|---|---|
| W1 | 5 |
| W2 | 5 |
| W3 | 4 |
| W4 | 6 |
| W5 | 2 |
| W6 | 6 |
| W7 | 4 |
| W8 | 3 |
| W9 | 5 |
| W10 | 5 |
| W11 | 4 |
| W12 | 6 |
| W13 | 5 |
| W14 | 4 |
| W15 | 3 |
| W16 | 3 |

W: Week.

season. The exclusion criteria resulted in a total of 15 players removed from the analysis. Three players were goalkeepers and thus had significantly different training and match demands; seven players missed more than seven consecutive days of training; three players were registered as part of the under-23 team; one player initiated the season in September (in-season started in July); one player participated in the national team.

From the 13 players examined, four were defenders, five were midfielders and four were attackers. Prior to data collection, participants were fully informed of the study design and signed a written informed consent. The study followed the ethical guidelines for human study as suggested by the Declaration of Helsinki. Furthermore, the study was approved by the research Ethics Committee of the Polytechnic Institute of Santarém, Santarém, Portugal (Nº24-2022ESDRM).

The small sample size is supported by previous studies in soccer [20–22]. Even so, the power of the sample size was calculated through G-Power [23]. Two Post-hoc analyses were conducted considering both types of the aim (comparisons and correlations). Thus, for the comparison analysis, an F-test, with a total of 13 participants with a $p < 0.05$ and effect-size of 0.2 and seven measurements (6 training sessions and 1 MD) was performed. The actual power achieved was 95.4%. Considering the correlation analysis, a Bivariate normal model with correlation of 0.7, $p < 0.05$ and the same 13 participants was applied which revealed 81.7% of actual power.

## Anthropometric and body composition collection

Body mass and height measures were obtained from the participants while dressed in light clothing without shoes using a stadiometer with an incorporated scale (Seca 220, Hamburg, Germany) according to standardised procedures [24]. Fat mass was collected from participants using the Inbody S10 (model JMW140, Biospace Co, Ltd., Seoul, Korea). The measurements were conducted in the morning in an ambient temperature and relative humidity of 22–23 ºC and 50–60%, respectively, after a minimum of 8-hr of fasting and following players' emptying their bladders. The participants did not exercise or ingest caffeine or alcohol during the 12-hr period prior to measurements. Recommendations from previous studies were followed to conduct all data collection procedures [25,26].

## Wellness quantification

The previous wellness questionnaire of McLean et al. [8] was applied individually 30-min before each training/match session through a specifically designed google form. The questionnaire uses a Likert scale ranging from 1–5 arbitrary units (A.U.), in which athletes were asked to rate their fatigue, sleep quality, muscle soreness, stress and mood (5 = very fresh, very restful, very great, very relaxed, and very positive mood, respectively; 1 = always tired, insomnia, very sore, highly stressed, and highly annoyed/irritable/down, respectively). All players were previously familiarised with the questionnaire during the previous season.

## Internal intensity quantification

The CR-10 Borg's scale [27] was also employed to monitor the participants rating of perceived exertion (RPE). Following 20- to 30-min post-session, every player provided a perceived exertion value using a google form specifically designed by answering the following question: "how intense was the training session?". The scale varied from 0 to 10 A.U., where each value rated as: 0 –nothing to all; 0.5 –extremely weak; 1 –very weak; 2 –weak; 3 –moderate; 4 –somewhat strong; 5 –strong; 7 –very strong; and 10 –extremely strong. Moreover, this scale has been previously used in several soccer studies [28]. In addition, the duration of the entire training

session and/or match in minutes was multiplied by the RPE to generate the session-RPE (A. U.) [29,30]. All players were already familiarised with the questionnaire from the previous season.

## External intensity quantification

Training and match load demands were measured using a 10 Hz GPS Vector S7 (Catapult Innovations, Melbourne, Australia). To avoid inter-unit bias, the same unit was worn by each player throughout the analysis period. The unit was placed on the upper back of the players 30-min before each session (training and match) and removed immediately following session completion.

The GPS device utilised has previously been validated for accuracy and reliability regarding measures of distance, velocity and average acceleration [31]. The following measures were used for analysis: (i) high-speed running distance (20–25 km/h) and sprint distance (>25 km/h) [32], number of accelerations ($> 2 m/s^2$) and number of decelerations ($< 2 m/s^2$) [33].

## Statistical analysis

Descriptive statistics are presented as mean ± standard deviation. Normality of the different variables was tested using the Shapiro-wilk test. Only the variables of quality sleep, session duration, high-speed running distance and deceleration presented normal distribution ($p > 0.05$). Thus, the intra-week variations (training sessions and matches) of these variables were analysed using the repeated measures ANOVA with the Bonferroni test for pairwise comparisons. The remaining variables were analysed through the Friedman ANOVA test with multiple comparisons. Significant results were considered at $p < 0.05$. The Hedges effect-size (ES) was performed to determine the effect magnitude through the difference of two means divided by the standard deviation from the data and the following criteria were used: $<0.2$ = trivial, 0.2 to 0.6 = small effect, 0.6 to 1.2 = moderate effect, 1.2 to 2.0 = large effect, and $>2.0$ = very large [34].

Finally, the relationship between wellness and intensity variables were explored using the Spearman's Rho correlation coefficient. The magnitude of correlations were classified as trivial (0.00 to 0.09), small (0.10 to 0.29), moderate (0.30 to 0.49), large (0.50 to 0.69), very large (0.70 to 0.89), and nearly perfect ($> 0.90$) [35]. All statistical procedures were executed in the IBM SPSS Statistics for Windows version 23.0 (IBM Corp Armonk, NY, USA).

## Results

Table 2 presents the wellness and load measures during training and matches while S1 Table presents all ES values. Sleep quality and fatigue were lowest on MD+1 compared with all other days with a very large ES. Moreover, fatigue showed higher values on MD compared with MD-4, MD-3 and MD-2 with trivial to small ES. Muscle soreness was lowest on MD+1 compared with all other days with a moderate to large ES. Muscle soreness showed higher values on MD compared with MD-4, MD-3, MD-2 and MD-1 with trivial to small ES. Additionally, stress showed to be lowest on MD+1 compared with all other days with small ESs. Finally, mood showed to be lower on MD+1 when compared with MD-5 with very large ES and also when compared with MD-4, MD-3, MD-2, MD-1 and MD with large ES.

Regarding session duration, MD+1 was significantly lower than MD-4 and MD with very large ESs. MD-5 was significantly lower than MD-4 and MD with very large ESs, while it was also significantly higher than on MD-2 and MD-1 with very large ESs. From MD-4 to MD-1, session duration significantly decreased with very large ESs. Finally, MD had the highest session duration when compared with all other days with very large ESs.

**Table 2. Descriptive statistics (mean ± standard deviation) of wellness and load demands.**

| Variables | MD+1 | MD-5 | MD-4 | MD-3 | MD-2 | MD-1 | MD |
|---|---|---|---|---|---|---|---|
| Quality of sleep (A.U.) | 3.2 ± 0.4 [a,b,c,d,e,f] | 4.0 ± 0.3 | 4.0 ± 0.2 | 4.0 ± 0.1 | 3.9 ± 0.2 | 4.0 ± 0.1 | 4.0 ± 0.2 |
| Fatigue (A.U.) | 2.8 ± 0.9 [a,b,c,d,e,f] | 3.7 ± 0.5 | 3.5 ± 0.5 [f] | 3.5 ± 0.5 [f] | 3.5 ± 0.5 [f] | 3.7 ± 0.4 | 3.9 ± 0.5 |
| Muscle Soreness (A.U.) | 2.9 ± 0.8 [a,b,e,f] | 3.5 ± 0.6 | 3.4 ± 0.5 [f] | 3.4 ± 0.5 [f] | 3.5 ± 0.5 [f] | 3.6 ± 0.5 [f] | 3.7 ± 0.5 |
| Stress (A.U.) | 3.4 ± 0.4 [a,b,c,d,e,f] | 3.7 ± 0.6 | 3.6 ± 0.5 | 3.6 ± 0.5 | 3.6 ± 0.6 | 3.7 ± 0.6 | 3.7 ± 0.6 |
| Mood (A.U.) | 3.3 ± 0.3 [a,b,c,d,e,f] | 3.7 ± 0.3 | 3.7 ± 0.5 | 3.6 ± 0.5 | 3.7 ± 0.5 | 3.7 ± 0.5 | 3.7 ± 0.5 |
| RPE (A.U.) | 3.8 ± 1.7 [b,f] | 5.5 ± 0.3 [b,c,d,e,f] | 6.6 ± 0.5 [d,e] | 6.3 ± 0.7 [d,e] | 4.1 ± 0.5 [e,f] | 2.6 ± 0.7 [f] | 7.3 ± 1.5 |
| Session duration (min) | 29.8 ± 10.4 [b,f] | 39.1 ± 1.6 [b,d,e,f] | 45.3 ± 3.5 [c,d,e,f] | 40.2 ± 1.8 [d,e,f] | 35.9 ± 1.9 [e,f] | 27.2 ± 1.7 [f] | 69.8 ± 21.8 |
| s-RPE (A.U.) | 150.1 ± 99.5 [b,c,f] | 225.1 ± 15.0 [b,c,d,e,f] | 302.0 ± 36.3 [c,d,e,f] | 263.6 ± 30.0 [d,e,f] | 156.9 ± 17.3 [e,f] | 73.7 ± 18.3 [f] | 570.1 ± 219.6 |
| HSR (m) | 94.9 ± 80.3 [b,f] | 114.6 ± 37.3 [b,c,d,e,f] | 206.7 ± 33.0 [d,e,f] | 200.9 ± 51.8 [d,e,f] | 47.1 ± 15.2 [e,f] | 18.7 ± 8.4 [f] | 422.5 ± 136.7 |
| Sprint distance (m) | 34.1 ± 30.7 [b,f] | 24.8 ± 15.5 [b,d,e,f] | 74.3 ± 17.5 [c,d,e] | 43.0 ± 18.0 [d,e,f] | 5.3 ± 3.2 [f] | 2.4 ± 1.8 [f] | 102.6 ± 42.8 |
| Accelerations (nr) | 24.0 ± 19.1 [a,b,c,f] | 46.4 ± 10.6 [d,e,f] | 47.6 ± 5.5 [d,e,f] | 49.7 ± 6.4 [d,e,f] | 32.7 ± 4.6 [e,f] | 17.9 ± 3.3 [f] | 79.9 ± 24.0 |
| Decelerations (nr) | 19.6 ± 16.7 [a,b,c,f] | 40.4 ± 9.9 [d,e,f] | 42.7 ± 5.8 [d,e,f] | 44.8 ± 6.0 [d,e,f] | 29.5 ± 4.4 [e,f] | 14.8 ± 2.1 [f] | 81.0 ± 25.0 |

MD: Match day; MD+1: One day after the match day); MD-5: Five days before match day; MD-4: Four days before match day; MD-3: Three days before match day; MD-2: Two days before match day; MD-1: One day before match day; RPE: Rate of perceived exertion using the CR-10 Borg's scale; Session-RPE: Multiplication of time of session by the score of RPE; A.U.: Arbitrary units; m: Meters; min: Minutes; nr, number; HSR: High speed running distance (20–25 km/h); significant different at MD-5[a]; MD-4[b]; MD-3[c]; MD-2[d]; MD-1[e]; MD[f]; at p<0.05.

Considering internal load measures of RPE and s-RPE, the results were similar. On the one hand, RPE showed that MD+1 had lower values than MD-4, MD-3 and MD with very large ESs. MD-5 had significantly lower values than MD-4, MD-3 and MD while it had significantly higher values than MD-2 and MD-1 (all with very large ESs). MD-4 and MD-3 had significantly higher values than MD-2 and MD-1 with very large ESs. MD-2 had also significantly higher values than MD-1 with very large ESs. Both MD-2 and MD-1 had significantly lower values than MD with very large ESs. On the other hand, s-RPE showed that MD+1 had significantly lower values than MD-4 and MD-3 with large to very large ESs. MD-5 had significantly lower values than MD-4 and MD-3 while it had significantly higher values than MD-2 and MD-1 (all with very large ESs). MD-4 and MD-3 had significantly higher values than MD-2 and MD-1 with very large ESs. MD-2 had significantly higher values than MD-1 with very large ES. Finally, MD had the highest value of the microcycle with very large ESs.

Regarding external load, high-speed running showed that MD+1 had significantly lower values than MD-4 and MD-3 with large to very large ESs. MD-5 had significantly lower values than MD-4 and MD-3 while it had significantly higher values than MD-2 and MD-1 (all with very large ESs). MD-4 and MD-3 had significantly higher values than MD-2 and MD-1 with very large ESs. MD-2 had significantly higher values than MD-1 with very large ES. Finally, MD had the highest values of the microcycle with very large ESs.

Sprinting showed significantly lower values on MD+1 than MD-4 and MD with very large ESs. MD-5 had significantly lower values than MD-4, MD-2 and MD-1 with very large ESs. MD-4 had significantly higher values than MD-3, MD-2 and MD-1 with very large ESs. MD-3 had significantly higher values than MD-2 and MD-1 with very large ESs. Finally, MD had significantly higher values when compared with all other days with very large ESs (with the exception of MD-4 where no significant differences were found).

Number of accelerations and deceleration presented a similar pattern. They showed significantly lower values on MD+1 than MD-5, MD-4 and MD-3 with very large ESs. They also showed higher values on MD-5, MD-4 and MD-3 than MD-2 and MD-1 with very large ESs,

respectively. They also showed higher values on MD-2 than MD-1 with very large ESs. Finally, MD showed higher values of the microcycle with very large ESs.

Table 3 presents the correlation coefficients among wellness and intensity variables using average values of the microcycle (including the match). Significant positive and very large correlations were found between fatigue and muscle soreness; stress and muscle soreness; stress and mood; and accelerations and decelerations. In addition, significant positive large correlations between muscle soreness and mood; RPE and s-RPE; session duration and s-RPE; high-speed running and sprint distance; high-speed running distance and accelerations; and sprint distance and accelerations. Finally, there was a significant negative large correlation between fatigue and s-RPE.

Table 4 presents the correlation coefficients between MD demands and wellness during MD+1. Significant small to very large negative correlations were found for all variables with the following exceptions: stress and mood did not show any correlation with any intensity variable; both quality of sleep and muscle soreness did not correlate with sprint distance.

Table 5 presents the correlation coefficients between MD demands and MD-1 wellness. Significant large negative correlations were found between the following variables: fatigue and session duration; fatigue and s-RPE; muscle soreness and session duration; muscle soreness and s-RPE; muscle soreness and deceleration.

## Discussion

The aims of this study were to: (i) analyse the within-microcycle variations in professional soccer players; (ii) analyse the relationships between wellness and training and match load demands; (iii) analyse the relationships between match-day (MD) demands and wellness during the following day (MD+1); and (iv) analyse the relationships between MD and wellness during the day before match-play (MD-1).

**Table 3. Correlation coefficient (r) between wellness and load demands.**

| Variables | Fatigue | Muscle Soreness | Stress | Mood | RPE | Session duration | s-RPE | HSR | Sprint distance | Acc | Dec |
|---|---|---|---|---|---|---|---|---|---|---|---|
| Quality of sleep | 0.363 | 0.467 | 0.396 | 0.396 | -0.126 | -0.110 | -0.198 | 0.313 | 0.077 | 0.126 | -0.044 |
| Fatigue | | **0.868 p<0.001** | 0.473 | 0.357 | -0.242 | -0.467 | **-0.593 p = 0.033** | 0.192 | 0.286 | 0.110 | 0.187 |
| Muscle Soreness | | | **0.780 p = 0.002** | **0.698 p = 0.008** | -0.165 | -0.418 | -0.418 | 0.269 | 0.170 | -0.011 | -0.044 |
| Stress | | | | **0.973 p<0.001** | -0.192 | -0.247 | -0.137 | 0.225 | -0.176 | -0.291 | -0.335 |
| Mood | | | | | -0.143 | -0.176 | -0.060 | 0.187 | -0.148 | -0.297 | -0.390 |
| RPE | | | | | | 0.077 | **0.648 p = 0.017** | 0.313 | 0.445 | 0.209 | 0.357 |
| Session duration | | | | | | | **0.681 p = 0.010** | -0.335 | -0.198 | 0.066 | 0.225 |
| s-RPE | | | | | | | | 0.060 | -0.055 | 0.104 | 0.280 |
| HSR | | | | | | | | | **0.588 p = 0.035** | **0.654 p = 0.015** | 0.484 |
| Sprint distance | | | | | | | | | | **0.593 p = 0.033** | 0.495 |
| Acc | | | | | | | | | | | **0.830 p<0.001** |

RPE: Rate of perceived exertion using the CR-10 Borg's scale; Session-RPE: Multiplication of session duration by the score of RPE; HSR: High speed running distance (20–25 km/h); Acc: Acceleration; Dec: Deceleration; Bold denotes significant correlations.

**Table 4. Correlation coefficient (r) between match demands and wellness during MD+1.**

| Variables | RPE | Session duration | s-RPE | HSR | Sprint distance | Acc | Dec |
|---|---|---|---|---|---|---|---|
| Quality of sleep | **-0.572**<br>**p = 0.041** | **-0.554**<br>**p = 0.050** | **-0.694**<br>**p = 0.008** | **-0.601**<br>**p = 0.030** | -0.331 | **-0.664**<br>**p = 0.013** | **-0.647**<br>**p = 0.017** |
| Fatigue | **-0.897**<br>**p<0.001** | **-0.698**<br>**p = 0.008** | **-0.846**<br>**p<0.001** | **-0.764**<br>**p = 0.002** | **-0.209**<br>**p = 0.494** | **-0.813**<br>**p = 0.001** | **-0.852**<br>**p<0.001** |
| Muscle Soreness | **-0.674**<br>**p = 0.012** | **-0.591**<br>**p = 0.033** | **-0.702**<br>**p = 0.008** | **-0.638**<br>**p = 0.019** | -0.118 | **-0.790**<br>**p = 0.001** | **-0.812**<br>**p = 0.001** |
| Stress | 0.084 | -0.393 | -0.221 | -0.105 | -0.512 | -0.290 | -0.127 |
| Mood | 0.220 | -0.155 | -0.017 | -0.033 | -0.274 | -0.113 | 0.000 |

RPE: Rate of perceived exertion using the CR-10 Borg's scale; Session-RPE: Multiplication of session duration by the score of RPE; HSR: High speed running distance (20–25 km/h); Acc: Acceleration; Dec: Deceleration; Bold denotes significant correlations.

Regarding aim (i), wellness variables remained similar across all training sessions, including MD, except for MD+1, where lower values were found suggesting lower wellness (for all variables). Considering internal and external measures, the lowest values were found on MD-1 while the highest were found on MD. Thus, the following pattern was revealed, MD+1 < MD-5 < MD-4 > MD-3 > MD-2 > MD-1 < MD. These findings have been confirmed in a recent systematic review on training intensity in professional soccer players that found the lowest values on MD-1 and higher values on MD-5, MD-4, and MD-3 [17]. Such a reduction on MD-1 is possibly associated with a tapering period intended to decrease fatigue and increase recovery in order to optimally prepare and perform in the following match [36].

Aim (ii) found some relationships between: fatigue and muscle soreness; stress and muscle soreness; muscle soreness and mood; stress and mood; RPE and s-RPE; session duration and s-RPE; accelerations and decelerations; high-speed running and sprint distance; high-speed running distance and accelerations; and sprint distance and accelerations. While finding associations among wellness and each internal and external variable, respectively, is quite understandable that the only major finding identified was between fatigue and s-RPE (large negative correlation). This may be associated with the fact that the study aim (ii) considered the average values of the microcycle (including the match). From a practical perspective for coaches and their staff, this approach does not seem optimal.

Considering that wellness measures collect data that is associated with the day before, it seems that match-play contributes to lower wellness markers. Such tendency was confirmed

**Table 5. Correlation coefficient (r) between match demands and MD-1 wellness.**

| Variables | RPE | Session duration | s-RPE | HSR | Sprint distance | Acc | Dec |
|---|---|---|---|---|---|---|---|
| Quality of sleep | 0.202 | -0.251 | -0.172 | 0.135 | -0.144 | -0.003 | -0.219 |
| Fatigue | -0.193 | **-0.634**<br>**p = 0.020** | **-0.609**<br>**p = 0.027** | -0.288 | -0.357 | -0.449 | -0.476 |
| Muscle Soreness | -0.373 | **-0.676**<br>**p = 0.011** | **-0.626**<br>**p = 0.022** | -0.299 | -0.285 | -0.468 | **-0.560**<br>**p = 0.047** |
| Stress | 0.062 | -0.265 | -0.150 | 0.003 | -0.110 | -0.205 | -0.202 |
| Mood | -0.046 | -0.256 | -0.153 | -0.171 | -0.409 | -0.279 | -0.199 |

RPE: Rate of perceived exertion using the CR-10 Borg's scale; Session-RPE: Multiplication of session duration by the score of RPE; HSR: High speed running distance (20–25 km/h); Acc: Acceleration; Dec: Deceleration; Bold denotes significant correlations.

by several correlations [Table 4, aim (iii)], namely, quality of sleep, fatigue and muscle soreness that correlated with RPE, s-RPE, high-speed running distance, accelerations and decelerations with large to very large magnitudes. In this sense, a similar result was also observed between s-RPE and time of sleep following MD in youth soccer players [15], which may partly explain the lower sleep quality and consequently a higher feeling of muscle soreness and fatigue reported in the present study. This finding was not observed in a study in youth soccer players that showed a tendency of higher external and internal intensities to be associated with improved sleep (quality and quantity), and feeling rested [14]. Notably, it is relevant to highlight, although expected, that MD session duration was higher when compared with all other training sessions, which also supports the previous assertion, namely, higher duration with higher intensity impairs wellness variables.

Contrastingly, a further study in youth soccer players found that high-intensity training had no impact on sleep quality and quantity [37]. While a study conducted with professional soccer players reported that sleep quality was not impacted by higher intensity sessions (MD included) [38]. Thus, more research is warranted to confirm the present findings.

Regarding the second aim of this study, Table 3 included average data of all sessions (including MD). The most relevant finding was the association between fatigue and s-RPE suggesting that higher values of fatigue were associated with lower values of internal intensity or vice-versa. However, when considering only the RPE, this relationship was not verified and thus it is necessary to determine that the RPE is associated with varying factors. For example, the RPE scores were responsive to hot vs. cold environments and elevated blood lactate concentrations resulting from repeated sprints or small-sided games in soccer players [39]. Thus, it still remains unclear whether the multiplication of the RPE by the training session duration presents a meaning in the opposite direction compared with only analysing the RPE of the session. Thus, s-RPE may not be closely linked with exercise duration [29,40].

When analysing match demands and MD-1 wellness (aim iv), the same relationship between fatigue and s-RPE was found and between fatigue and session duration which may suggest that lower fatigue values (which suggests greater fatigue) were associated with higher session duration. These findings were congruent with those reported in Table 4. Furthermore, muscle soreness showed a similar association with session duration, s-RPE and decelerations, which reinforces the previous analysis (aim iii). Therefore, it is relevant to highlight that decelerations were only associated with muscle soreness and no other wellness variable. Previously, the s-RPE has been found to be moderately correlated with fatigue and muscle soreness [41], which is in line with the findings of the current study.

There is scarce literature on the previous relationships (both aims iii and iv), however according to the results of the present study, it seems that higher internal and external intensity contributed to a lower wellness state, especially MD intensity was associated with wellness variables on MD+1 (except for mood and stress). Similar findings were not found in professional soccer players that showed a relationship between higher external training intensity and the following sleep night [42] or between s-RPE and sleep quality [43]. In support, training monotony of s-RPE was correlated with accumulated sleep quality over the season in youth soccer players [13]. However, such metrics were not analysed in this study, although there is speculation that these metrics would show similar associations between sleep quality and external measures in a different direction when compared to the present study.

In contrast, a recent study in youth soccer players that analysed similar relationships across the weeks (using weekly average data), suggested that with higher values of RPE and s-RPE, higher levels of muscle soreness and fatigue may occur and improved readiness and sleep quality [14]. This first association is similar with the present study findings while the second is

dissimilar. However, the design of the present study used data from each session and specifically MD data in comparison to the rest of the microcycle.

The present study had some limitations that need to be addressed. Namely, the small sample size from only one team and the analysis period of only 16-weeks. Consequently, future studies should attempt to analyse larger sample sizes, include more pre- and in-season periods (e.g. pre-season, early-season, mid-season and end-season), and analyse regular weeks with one match versus congested weeks with more than one match as previously suggested [14]. Future studies that consider large sample sizes and number of teams should also conduct some regression analysis and analyse other contextual variables such as the result of the match as previously reported. Namely that a win may contribute to better sleep quality when compared to other results [44]. Another contextual variable to consider is match location as previous research showed that away matches that required longer travel distance tended to decrease sleep quality and wake behaviors [45], which may consequently decrease other wellness variables.

Nonetheless, the majority of the correlations were large and very large and thus should be considered by coaches. Future research should aim to reinforce the previous recommendation regarding the continuous monitorisation of key metrics [15]. Finally, it is also suggested to replicate this study design in other league teams, with female players and different sports while avoiding the limitations listed.

## Conclusions

As hypothesised, internal and external intensity and wellness varied across the microcycle. Specifically, MD was the most demanding session of the week while MD-1 was the lowest load for both internal and external variables with very large ESs. Wellness only showed a variation on MD+1 when compared to all other training sessions. In this case, wellness revealed lower values with very large ESs which was associated with worse sleep quality and mood, while higher levels of stress, fatigue and muscle soreness were also observed.

In line with previous findings, the second hypothesis was also confirmed as several relationships were found between intensity and wellness variables. In this regard, a tendency of higher internal and external intensity (of matches) was associated with lower wellness, specifically sleep quality, muscle soreness and fatigue on MD+1 (small to very large correlations). A similar association between match demands and wellness (collected on the same day) were also reported (large correlations). Specifically, some relationships were found between fatigue and s-RPE as well as between fatigue and session duration, suggesting that greater fatigue was associated with higher s-RPE and higher session duration. Moreover, muscle soreness highlighted the same association between session duration, s-RPE and decelerations. When considering the average microcycle data between intensity and wellness variables, only fatigue was negatively associated with s-RPE.

The findings of the present study suggest that internal and external MD load is the highest of the microcycle and consequently, there is a tendency for lower wellness during the other training days of the week, while wellness remains constant. Such information should be confirmed in future research. Still, this study suggests that coaches and their staff should carefully pay attention to the wellness measures obtained in the post-MD period to better adjust load in the recovery days following the match. Considering the changes in internal and external load measures across the microcycle, it seems that apart from MD+1, wellness may be better managed during the remaining days. Thus, the importance of constant monitoring is relevant to implement improved, tailored recovery strategies in the days following MD, which seems to be evident from the present study considering the obtained results.

## Supporting information

**S1 Table. Effect sizes of the comparisons presented in Table 2.** MD: Match day; MD+1: One day after the match day); MD-5: Five days before match day; MD-4: Four days before match day; MD-3: Three days before match day; MD-2: Two days before match day; MD-1: One day before match day; RPE: Rate of perceived exertion using the CR-10 Borg's scale; Session-RPE: Multiplication of time of session by the score of RPE; A.U.: Arbitrary units; m: Meters; min: Minutes; nr, number; HSR: High speed running distance (20–25 km/h).
(DOCX)

## Acknowledgments

The authors would like to thank the team's coaches and players for their cooperation during all data collection procedures.

## Author Contributions

**Conceptualization:** Rafael Franco Soares Oliveira, Rui Canário-Lemos, João Paulo Brito.

**Data curation:** Rafael Franco Soares Oliveira, Rui Canário-Lemos.

**Formal analysis:** Rafael Franco Soares Oliveira.

**Funding acquisition:** Rafael Franco Soares Oliveira.

**Investigation:** Rafael Franco Soares Oliveira, Rui Canário-Lemos, Rafael Peixoto, José Vilaça-Alves, Ryland Morgans, João Paulo Brito.

**Methodology:** Rafael Franco Soares Oliveira, Rui Canário-Lemos, João Paulo Brito.

**Project administration:** Rafael Franco Soares Oliveira, Rui Canário-Lemos, João Paulo Brito.

**Resources:** Rui Canário-Lemos.

**Software:** Rafael Franco Soares Oliveira, Rui Canário-Lemos.

**Supervision:** Rafael Franco Soares Oliveira, Rui Canário-Lemos, João Paulo Brito.

**Validation:** Rafael Franco Soares Oliveira, João Paulo Brito.

**Visualization:** Rafael Franco Soares Oliveira, Rui Canário-Lemos.

**Writing – original draft:** Rafael Franco Soares Oliveira, Rui Canário-Lemos, Rafael Peixoto, José Vilaça-Alves, Ryland Morgans, João Paulo Brito.

**Writing – review & editing:** Rafael Franco Soares Oliveira, Rui Canário-Lemos, Rafael Peixoto, José Vilaça-Alves, Ryland Morgans, João Paulo Brito.

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
