## [Decision Letter · Decision Letter 0]

7 Jun 2023

PONE-D-23-04228Relationships between wellness, internal and external intensity measures from a European professional male soccer teamPLOS ONE

Dear Dr. Oliveira,

Thank you for submitting your manuscript to PLOS ONE. After careful consideration, we feel that it has merit but does not fully meet PLOS ONE’s publication criteria as it currently stands. Therefore, we invite you to submit a revised version of the manuscript that addresses the points raised during the review process.

We look forward to receiving your revised manuscript.

Kind regards,

Bruno Emanuel Nogueira Figueira

Academic Editor

PLOS ONE

Journal Requirements:

Whilst you may use any professional scientific editing service of your choice, PLOS has partnered with both American Journal Experts (AJE) and Editage to provide discounted services to PLOS authors. Both organizations have experience helping authors meet PLOS guidelines and can provide language editing, translation, manuscript formatting, and figure formatting to ensure your manuscript meets our submission guidelines. To take advantage of our partnership with AJE, visit the AJE website (http://learn.aje.com/plos/) for a 15% discount off AJE services. To take advantage of our partnership with Editage, visit the Editage website (www.editage.com) and enter referral code PLOSEDIT for a 15% discount off Editage services.  

If the PLOS editorial team finds any language issues in text that either AJE or Editage has edited, the service provider will re-edit the text for free.

a) The name of the colleague or the details of the professional service that edited your manuscript.

b) A copy of your manuscript showing your changes by either highlighting them or using track changes (uploaded as a *supporting information* file).

c) A clean copy of the edited manuscript (uploaded as the new *manuscript* file).

"This research was funded by the Portuguese Foundation for Science and Technology, I.P., Grant/Award Number UIDP/04748/2020. The study was conducted according to the guidelines of the Declaration of Helsinki and approved by the Ethics Committee of Polytechnic Institute of Santarém (Nº24-2022ESDRM) The authors declare no conflict of interest. The funders had no role in the design of the study.."

"This research was funded by the Portuguese Foundation for Science and Technology, I.P., Grant/Award Number UIDP/04748/2020. The funders had no role in the design of the study; in the collection, analyses, or interpretation of data; in the writing of the manuscript, or in the decision to publish the results."

6. We note that you have indicated that data from this study are available upon request. PLOS only allows data to be available upon request if there are legal or ethical restrictions on sharing data publicly. For more information on unacceptable data access restrictions, please see http://journals.plos.org/plosone/s/data-availability#loc-unacceptable-data-access-restrictions. 

7. Your ethics statement should only appear in the Methods section of your manuscript. If your ethics statement is written in any section besides the Methods, please delete it from any other section. 

Reviewers' comments:

Reviewer's Responses to Questions

**Comments to the Author**

1. Is the manuscript technically sound, and do the data support the conclusions?

Reviewer #1: Yes

Reviewer #2: Yes

Reviewer #3: Yes

2. Has the statistical analysis been performed appropriately and rigorously? 

Reviewer #1: Yes

Reviewer #2: Yes

Reviewer #3: Yes

3. Have the authors made all data underlying the findings in their manuscript fully available?

Reviewer #1: No

Reviewer #2: Yes

Reviewer #3: Yes

4. Is the manuscript presented in an intelligible fashion and written in standard English?

Reviewer #1: Yes

Reviewer #2: Yes

Reviewer #3: Yes

5. Review Comments to the Author

Reviewer #1: Dear authors:

Overall, the study aimed to investigate within-microcycle variations of professional soccer players, analyzing the relationships between wellness and load demands, as well as the relationships between match demands and wellness of MD+1 and MD-1. The results showed that wellness variables remained similar across all sessions, except for MD+1 where lower values were found. The study also found that the lowest internal and external intensity values were found on MD-1 while the highest were found on MD. These findings are consistent with previous studies that have also found lower values on MD-1 and higher values on MD-5, MD-4, and MD-3.

The authors suggest that the reduction in wellness on MD-1 may be associated with a tapering period aimed at decreasing fatigue and increasing recovery to achieve better performance in the following match. The authors also suggest that the results of this study may help coaches and practitioners to better plan and monitor training loads and recovery strategies for professional soccer players.

However, it should be noted that the study has some limitations, such as the small sample size and the fact that only one team was studied. Therefore, further research with larger samples and multiple teams is needed to confirm the findings of this study. Additionally, future studies could investigate the impact of different tapering strategies on player performance and wellness. In the below you can find my comments to improve some sections:

Title

Based on the content of the study, a possible suggestion for the title could be "Relationships between internal and external load demands and wellness in professional soccer players across microcycles".

Introduction

It is well written. There is no comment in this section.

Method:

Participant section:

Overall, the participant section is well written and provides relevant information about the sample size, characteristics of the players, and inclusion criteria. However, as a reviewer, I would suggest adding more information about the recruitment process and how participants were selected for the study. It would also be useful to know if any players were excluded from the study and for what reasons.

Additionally, the authors could provide more information on how they ensured compliance with the eligibility criteria, such as how they tracked participation in training sessions and wellness and training intensity reports. This information would increase the transparency of the study and help readers understand the validity of the data collected.

Finally, the authors should clarify whether the power calculations were conducted prior to or after data collection. If it was done prior to data collection, it would be useful to know how the authors arrived at the effect-size of 0.2 and the correlation coefficient of 0.7 used in the calculations.

External intensity quantification & Internal intensity quantification section:

The methods section regarding the external and internal intensity quantification is well described and adequately detailed. However, it could be improved by providing more information on the interpretation of the results obtained through the GPS Vector S7. For instance, the authors could explain how the measured variables, such as high-speed running and sprinting, relate to the overall physical performance of the players. Moreover, it would be helpful to mention the standard deviation or range of values for each variable to provide a better understanding of the players' physical demands during training and matches. Finally, the authors could also provide information on how the CR-10 Borg’s scale is typically used to assess RPE in soccer players, and how session-RPE is calculated based on this scale.

Discussion

As a general view, here are a few suggestions for improving the conclusions section:

Provide more context: While the conclusions do summarize the findings of the study, they could benefit from additional context. It would be helpful to briefly explain why the study findings are significant and what implications they have for future research or practice.

Be more specific: The conclusions mention several associations between intensity and wellness variables, but they do not provide specific details or effect sizes. Including more specific information would help readers better understand the significance of these relationships.

Discuss future directions: Given the findings of the study, it would be valuable to suggest areas for future research or potential interventions that could be developed to address the identified relationships between intensity and wellness variables.

Reviewer #2: Thank you to the authors for putting together this comprehensive manuscript. It is very interesting with very relevant and practical findings; however, the quality of writing needs to be improved significantly. See some comments below:

Abstract

• Indicate which measures are IL and which are EL

• Change ap-plied to “applied”

• Rephrase to: “While MD-1 presented the lowest internal (variables) and external load (variables) values, MD presented the highest values”

• When you say “time of the session” – are you referring to the time of day or session duration?

Intro

• Line 2: “…is a common practice to reduce to the risk of injury and overtraining.”

• Line 3: “…practitioners employ periodization strategies by adjusting training intensities considering the previous sessions.”

• Line 14: strengthen topic sentence of this paragraph -

• Line 15: begin sentence with “specifically”

• Paragraph 2 needs to be strengthened with improved writing quality

• Line 29: add a comma after session and after wellness perception

• Line 32: briefly note what the relationship outcome was “This was evident in a recent study on youth soccer players, where sleep and intensity…” � discuss in that one sentence was the main outcome was

Methods

• Design – you only mention sleep, but you monitored more wellness variables also. Either list them all here or introduce all in the later section

• Rephase: “..before each training/match session through a specifically designed google form.”

• The questionnaire uses a Likert scale ranging from 1-5, in which athletes were asked to rate their fatigue, sleep quality, muscle soreness, stress, and mood (5=very fresh, very restful, very great, very relaxed, and very positive mood, respectively; 1=always tired, insomnia, very sore, highly stressed, and highly annoyed/irritable/down, respectively). All players were already familiarized with the questionnaire from the previous season.”

• Combine lines 88 and 92 – very repetitive, just throw in the 10hz sampling session in line 88

• Lines 96-97: delete the authors from the text and just include their reference as validation

• Line 106: delete this topic sentence

• Line 108: change during to from

Results

• Table 2: add nr in the legend

• Lines 130-132: when you say wellness variables including sleep quality and fatigue, are those the only 2 wellness variables? Or are there others? This is unclear by the phrasing of the sentence. Please rephrase to be more clear

• Line 132: didn’t you just say this in the previous sentence?

• Line 133-134: again, repeating the last 2 sentences you wrote – synthesize, as this is very repetitive

• Even though all numbers are displayed in Table 2, please still include actual numbers in the results section. Don’t need every single number, but some means or p values or effect sizes would strengthen this section

• Please rephrase “time of session” to “session duration” throughout your paper

• This reads moreso like a list to me – I think by adding numbers as I suggested above will help break that up.

• For tables 4 and 5 results, why not run a regression for highly correlated variables?

Discussion

• Line 226: which wellness variables were lower in MD+1?

• If citing an SR, discuss/cite the studies individually

• Appears that discussion is out of order – you discuss aim 3 before aim 2

• Significant improvements in writing is needed

Reviewer #3: Congratulations to the authors. But some minor adjustments need to be made;

- Hypotheses should be written more clearly and precisely.

-Line 46. “This was an observational study design” please explain and show references.

-Line 46-47, sleep quality and duration?? You can use welness measures instead of sleep quality and duration.

Line- 48, give more details about the content of the 70 training sessions?

Line 49-50, please give more details about the reseacrh design? Which microcycles were chosen?

-In the Methods section, Please explain which sampling method was used to create my sample group. It is based on scientific source.

-The data collection process cannot be detailed in method section. Where, by whom, and in what time of day the data were collected. It should be detailed.

-Please give more details about eligibility criteria.

Line 61, a European soccer team or its country? Which league?

Line 59-60, body mass, height, and fat mass? How did you determine this? Please add fat free mass also. Please add the explanations in methods section. Were these measurements made on an empty stomach or on a full stomach? What time of day was it done? Were the participants informed about the criteria recommended by the ACSM before the measurements? Show references for these all measurements and explanations.

-In the resuts section, there are sentences that repeat each other. Please revise them.

-Please develop practical applications section of the study.

6. PLOS authors have the option to publish the peer review history of their article (what does this mean?). If published, this will include your full peer review and any attached files.

Reviewer #1: No

Reviewer #2: No

Reviewer #3: **Yes: **halil ibrahim ceylan

---

## [Author Response · Author response to Decision Letter 0]

22 Jun 2023

Dear Editor, please see the "cover letter", "Response to reviewers" and the "revised manuscript with track changes". 

Thank you

---

## [Decision Letter · Decision Letter 1]

12 Jul 2023

PONE-D-23-04228R1The relationship between wellness and training and match load in professional male soccer playersPLOS ONE

Dear Dr. Oliveira,

Thank you for submitting your manuscript to PLOS ONE. After careful consideration, we feel that it has merit but does not fully meet PLOS ONE’s publication criteria as it currently stands. Therefore, we invite you to submit a revised version of the manuscript that addresses the points raised during the review process.

We look forward to receiving your revised manuscript.

Kind regards,

Bruno Emanuel Nogueira Figueira

Academic Editor

PLOS ONE

Journal Requirements:

Additional Editor Comments (if provided):

Dear authors, thanks for the revisions made addressing both reviewers' comments. After carefully checking changes and improvements I have noticed some minor issues that should be implemented in a minor revision. Please, I encourage to revise the manuscript according to the following minor issues:

- L142-143, please cite the SPSS software following the company criterion: https://www.ibm.com/support/pages/how-cite-ibm-spss-statistics-or-earlier-versions-spss

- Consistency is required when using abbreviations (e.g., effect size: ES). Most of the times, the authors used “effect size” and a few times the abbreviation HA. If the intention is to use the abbreviation, on the first occasion write in full “effect size”, indicating the abbreviation in parentheses (ES). Then, always use ES.

- In table 1 abbreviations should appear using the presentation of tables 2 and 3, W: week.

- Provide DOI to all references: e.g. Reference 15, 22 and 23

Reviewers' comments:

Reviewer's Responses to Questions

**Comments to the Author**

1. If the authors have adequately addressed your comments raised in a previous round of review and you feel that this manuscript is now acceptable for publication, you may indicate that here to bypass the “Comments to the Author” section, enter your conflict of interest statement in the “Confidential to Editor” section, and submit your "Accept" recommendation.

Reviewer #1: All comments have been addressed

Reviewer #2: All comments have been addressed

Reviewer #3: All comments have been addressed

2. Is the manuscript technically sound, and do the data support the conclusions?

Reviewer #1: Yes

Reviewer #2: Yes

Reviewer #3: Yes

3. Has the statistical analysis been performed appropriately and rigorously? 

Reviewer #1: Yes

Reviewer #2: Yes

Reviewer #3: Yes

4. Have the authors made all data underlying the findings in their manuscript fully available?

Reviewer #1: Yes

Reviewer #2: (No Response)

Reviewer #3: Yes

5. Is the manuscript presented in an intelligible fashion and written in standard English?

Reviewer #1: Yes

Reviewer #2: Yes

Reviewer #3: Yes

6. Review Comments to the Author

Reviewer #1: Based on the comments, the authors have made the necessary corrections in the article. Congratulations to the authors for their valuable work.

Reviewer #2: Thank you for your updates to this paper. My remaining comment is for the results section, particularly for table 2. In the written text you note effect sizes, but there is no data on effect sizes in the table. Please include these numbers in the written text.

Reviewer #3: All corrections were made by Authors..my decision is the article can be published in current form. Congratulations to all authors...

7. PLOS authors have the option to publish the peer review history of their article (what does this mean?). If published, this will include your full peer review and any attached files.

Reviewer #1: No

Reviewer #2: No

Reviewer #3: **Yes: **Halil ibrahim ceylan

---

## [Author Response · Author response to Decision Letter 1]

13 Jul 2023

Dear Editor, 

Please find enclosure the revision of our manuscript: “The relationship between wellness and training and match load in professional male soccer players”. The revision includes three files as requested: a rebuttal letter that responds to each point raised by the academic editor and reviewer(s); a 'Revised Manuscript with Track Changes'; and an unmarked version of your revised paper without tracked changes labeled as 'Manuscript'. The answers presented in the rebuttal letter can also be find in the following text:

Comments of the Editor

Dear authors, thanks for the revisions made addressing both reviewers' comments. After carefully checking changes and improvements I have noticed some minor issues that should be implemented in a minor revision. Please, I encourage to revise the manuscript according to the following minor issues:

Answer: Dear editor, thank you very much for your suggestions which improve quality to our work. 

- L142-143, please cite the SPSS software following the company criterion: https://www.ibm.com/support/pages/how-cite-ibm-spss-statistics-or-earlier-versions-spss

Answer: The citation was updated as follows: “IBM SPSS Statistics for Windows version 23.0 (IBM Corp, Armonk, NY,USA).”

- Consistency is required when using abbreviations (e.g., effect size: ES). Most of the times, the authors used “effect size” and a few times the abbreviation HA. If the intention is to use the abbreviation, on the first occasion write in full “effect size”, indicating the abbreviation in parentheses (ES). Then, always use ES.

Answer: Done. Thank you for the attention. 

- In table 1 abbreviations should appear using the presentation of tables 2 and 3, W: week.

Answer: Done. 

- Provide DOI to all references: e.g. Reference 15, 22 and 23

Answer: DOI’s were added accordingly. 

Comments of Reviewer 2

Thank you for your updates to this paper. My remaining comment is for the results section, particularly for table 2. In the written text you note effect sizes, but there is no data on effect sizes in the table. Please include these numbers in the written text.

Answer: Dear reviewer, thank you for the positive feedback. Considering that adding the exact numbers of the ES would difficult the reading, we added a new table as a Supplementary table 1 to provide all ES values. Thank you.

Best regards

---

## [Editor Report · Decision Letter 2]

18 Jul 2023

The relationship between wellness and training and match load in professional male soccer players

PONE-D-23-04228R2

Dear Dr. Oliveira,

We’re pleased to inform you that your manuscript has been judged scientifically suitable for publication and will be formally accepted for publication once it meets all outstanding technical requirements.

Kind regards,

Bruno Emanuel Nogueira Figueira

Academic Editor

PLOS ONE
---

## [Editor Report · Acceptance letter]

21 Jul 2023

PONE-D-23-04228R2 

The relationship between wellness and training and match load in professional male soccer players 

Dear Dr. Oliveira:

I'm pleased to inform you that your manuscript has been deemed suitable for publication in PLOS ONE. Congratulations! Your manuscript is now with our production department. 

Kind regards, 

on behalf of

Dr. Bruno Emanuel Nogueira Figueira 

Academic Editor

PLOS ONE